# Engagement of primary care physicians in medication decision-making for patients with multimorbidity in China: A cross-sectional study

Liyan Han[1,2☯], Leyi Jiang[1☯], Mi Yao[3], Yu Xia[4], Ming Yan[5], Junhai Zhen[1], Lingyan Wu[1], Yi Guo[1], Yuling Tong[1]*, Zhijie Xu[1]*

1 Department of General Practice, The Second Affiliated Hospital, Zhejiang University School of Medicine, Hangzhou, China, 2 Zhangqi Township Health Center of Cixi City, Ningbo, China, 3 Department of General Practice, Peking University First Hospital, Beijing, China, 4 Department of General Practice, Peking University Shenzhen Hospital, Shenzhen, China, 5 Department of Emergency Medicine, Sir Run Run Shaw Hospital, Zhejiang University School of Medicine, Hangzhou, China

☯ These authors contributed equally to this work.
* zhijiexu@zju.edu.cn (ZX); tongyl0313@zju.edu.cn (YT)

## Abstract

### Background

Managing medications for patients with multimorbidity is complex and often leads to medication-related problems. Primary care physicians (PCPs) play a critical role in optimizing medication use but face considerable challenges and research on their practices remains limited. This study aimed to examine PCPs' medication decision-making practices for patients with multimorbidity, identify associated factors, and explore perceived difficulties.

### Methods

A cross-sectional survey was conducted among PCPs in five regions of Zhejiang, China, using a self-administered questionnaire. Descriptive statistics were used to assess PCPs' engagement in medication decision-making and their perceived difficulties. Linear regression analyses were performed to identify factors associated with PCPs' overall engagement of medication decision-making practices.

### Results

A total of 346 valid responses were included in the final analysis. PCPs showed limited engagement in eliciting patient preferences and addressing individual concerns. Perceived difficulties included difficulties in understanding complex medication regimens and adverse reactions, evaluating overall benefits and risks, and discussing treatment alternatives. Linear regression analysis revealed that daily outpatient volume (B = −0.046, 95% CI: −0.083 to −0.008), prior training in medication use (B = 5.764, 95% CI: 1.559 to 9.969), and collaboration with pharmacists (B = 7.048,

**Data availability statement:** All relevant data are within the paper and its Supporting Information files.

**Funding:** The author(s) received no specific funding for this work.

**Competing interests:** The authors have declared that no competing interests exist.

95% CI: 3.313 to 10.782) were significantly associated with more appropriate decision-making practices.

## Conclusion

While many PCPs reported general engagement in medication decision-making, gaps remain in managing complex therapies and delivering patient-centered care. Targeted pharmaceutical training and strengthened interprofessional collaboration may support PCPs in improving medication decisions for patients with multimorbidity.

---

## Introduction

Multimorbidity, defined as the coexistence of two or more chronic conditions in an individual, is becoming increasingly prevalent worldwide [1]. It is estimated that approximately one-third of the global adult population lives with multimorbidity [2]. This condition often necessitates the use of multiple medications, which in turn increases the risk of medication-related problems [3,4]. These include drug-drug interactions, adverse drug reactions, medication non-adherence, and prescribing errors [4]. Such complications can result in serious consequences, including higher rates of hospitalization and mortality, increased healthcare costs and treatment burdens, and a diminished quality of life for patients [5,6].

Primary care physicians (PCPs) are uniquely positioned to optimize medication use among patients with multimorbidity. As the first point of contact in the healthcare system, PCPs are responsible for delivering continuous, comprehensive, and coordinated care [7]. They are well-placed to conduct comprehensive assessments, reconcile complex medication regimens, and tailor treatment plans based on patients' overall health status, preferences, and social contexts [8]. Through regular follow-ups and long-term patient relationships, PCPs can identify medication-related problems and adjust therapies accordingly (e.g., prescribing and deprescribing medications), making them pivotal in improving medication safety and therapeutic outcomes.

In China, this issue warrants particular attention. The primary health care (PHC) system in China has undergone significant reforms in recent years, with national policies emphasizing the importance of strengthening community-based care and improving medication safety [9,10]. PHC facilities have become the main site for chronic disease management, and are now responsible for caring for a large and growing population of patients with multimorbidity, estimated to account for 58% of all patients in some regions [2]. These changes are closely aligned with China's hierarchical medical system reform, particularly the implementation of the tiered diagnosis and treatment policy and the promotion of the family doctor contract service model [11]. In this context, PCPs bear the primary responsibility for medication management, including initiating, adjusting, and monitoring treatments across multiple conditions.

However, the capacity and willingness of PCPs to manage the complex medication needs of patients with multimorbidity remain uncertain. Numerous barriers hinder

their ability to optimize treatment, including clinical inertia, patient resistance to changes in medication regimens, limited access to decision support tools, inadequate training in managing polypharmacy, and time constraints during consultations [12,13]. Specifically, clinical inertia among PCPs often stems from their hesitancy to modify specialist-initiated regimens due to deference to authority, uncertainty about prescribing rationale, and legal concerns, thus compromising medication safety and care quality [14]. Moreover, the lack of clinical guidelines and insufficient evidence for treating patients with multimorbidity make it difficult to formulate individualized medication plans [15]. Most existing guidelines are disease-specific and do not address the competing priorities involved in managing coexisting conditions, further complicating clinical decision-making in primary care [16].

In our previous qualitative study, we found that PCPs face complexities and difficulties when making medication decisions for patients with multimorbidity. These difficulties arise from the inherently multifaceted nature of clinical decision-making, as well as the interaction of various factors such as drug–disease interactions, cognitive biases, and patient non-adherence driven by high treatment burdens [17]. For instance, PCPs often struggle to judge whether a medication prescribed for one condition may exacerbate another, or how to accurately weigh the overall benefit–risk profile of a treatment in the context of multiple coexisting illnesses. However, despite growing recognition of these challenges, empirical research remains limited. It is still unclear to what extent PCPs engage in various aspects of medication decision-making in daily practice, and what factors may influence their practices.

The aim of this study is to investigate the medication decision-making practices of PCPs in managing patients with multimorbidity, identify factors associated with their decision-making behavior, and explore their perceived challenges. By providing empirical evidence from real-world primary care settings, this study seeks to fill an important knowledge gap and inform the development of targeted interventions to improve the quality of medication management for patients with multimorbidity in China and similar healthcare contexts.

## Methods

### Study design, population and setting

This study employed a cross-sectional survey design. Data collection was conducted between December 11 and December 25, 2024, across 23 PHC facilities in five regions of Zhejiang Province, China: Hangzhou, Cixi, Daishan, Suichang, and Kaihua. Eligible participants were licensed physicians registered in general practice or internal medicine, with over one year of clinical experience in their current PHC facility. Physicians who declined to provide informed consent were excluded from the study.

A mixed sampling strategy combining convenience and purposive sampling was used to achieve both broad participation and targeted representation of key demographic groups relevant to the study objectives. The required sample size was calculated based on a ratio of 5–10 participants per survey item [18]. With an estimated 30 items in the final questionnaire and a projected 20% non-response rate, the target sample size was determined to be approximately 375 participants.

This study involves human participants and was approved by the Second Affiliated Hospital of Zhejiang University School of Medicine Ethics Committee [approval no. (2024) Ethics Review Research no. (1485)], and the research was conducted in accordance with the principles outlined in the Declaration of Helsinki [19].

### Instrument

The questionnaire used in this study was self-developed. The questionnaire used in the survey can be found in S1 File. The initial item pool was generated based on a comprehensive literature review of multimorbidity management, medication review, deprescribing, and clinical decision-making frameworks [1–6,20–22]. Drawing from this review, the research team developed the Medication Decision-Making for Multimorbidity Framework (MDMF),

which outlines a five-step process to guide PCPs in making appropriate medication decisions: (a) review of health problems, (b) comprehensive medication assessment, (c) shared decision-making, (d) documentation of medication therapy, and (e) follow-up arrangement [23]. Questionnaire items were designed to reflect these five domains.

A two-round Delphi consultation was conducted with 13 experts in general practice, clinical pharmacy, and medical ethics to refine the content and ensure face and content validity of the instrument (item-level CVI ranging from 0.79–1.00 and scale-level CVI averaging 0.88) [24]. Following discussions within the research team, the final version of the questionnaire was finalized. Prior to the formal study, the questionnaire underwent pilot testing with 112 PCPs in Hangzhou and Cixi to assess its psychometric properties. The instrument demonstrated good internal consistency (Cronbach's $\alpha = 0.87$ for the full scale), strong test–retest reliability (intraclass correlation coefficient = 0.84 over a two-week interval) and satisfactory construct validity ($\chi^2/df = 1.98$, CFI = 0.93, TLI = 0.91, RMSEA = 0.06) [18].

The questionnaire consisted of three main sections. The first section collected demographic and contextual information, including age, sex, educational background, practice location, and average daily outpatient volume, etc. The second section included 20 items assessing PCPs' engagement in medication decision-making for patients with multimorbidity over the past month. Items addressed practices such as reviewing medication regimens, discussing benefits and risks with patients, and incorporating patient preferences. Responses were rated on a three-point scale: "always", "occasionally", or "rarely". The third section contained 8 items measuring perceived challenges in medication decision-making, such as difficulties in evaluating drug–drug or drug–disease interactions. Respondents rated their agreement using a four-point Likert scale ranging from "strongly disagree" to "strongly agree".

## Data collection

The questionnaire was developed and uploaded to the online questionnaire platform (wenjuanxing, version 3.0.10, https://www.wjx.cn/), a survey tool known for its user-friendly interface and efficiency, enabling survey completion across a variety of digital devices, including smartphones, tablets, and computers. The content of the questionnaire was entered into the platform to generate an electronic version, which was then reviewed and refined by the research team to ensure accuracy. Once finalized, a survey link was generated and can be distributed to participants. Each PHC facility appointed a designated individual (typically a health administrator) to oversee the survey distribution. This person was responsible for explaining the purpose, significance, and content of the study to the participants. If participants had any questions, they were instructed to consult the designated contact person, who would then relay inquiries to the researchers for prompt clarification.

Written informed consent was obtained electronically on the first page of the questionnaire. Participants were provided with comprehensive information about the study, including details about the research team, the purpose of the study, the contents of the survey, and their rights and responsibilities. Participation in the survey was voluntary, and participants were free to withdraw at any point before submission. Confidentiality of data and anonymity of participants were strictly maintained throughout the study. To encourage active participation, a monetary incentive of 10 RMB was offered to each respondent who fully completed the questionnaire.

To ensure the quality and completeness of the responses, the survey incorporated built-in quality control measures [24]. The survey was designed to require full engagement from respondents, thus reducing the likelihood of incomplete responses. Each respondent, identified by a unique IP address, was allowed to submit the questionnaire only once. Upon submission, data from each completed questionnaire were automatically collected and transferred to a centralized web-based database [25]. After the survey concluded, the data were downloaded from the platform and manually reviewed. Questionnaires with unusually long or short response times, repetitive answer patterns (e.g., all answers selected as the same option), or inconsistencies (e.g., a 45-year-old physician with only 2 years of experience) were considered invalid and excluded from the analysis.

 

## Statistical analysis

Descriptive statistics were used to summarize participants' characteristics, their engagement in various aspects of medication decision-making, and the difficulties they perceived. These results were presented as frequencies and proportions to provide an overview of the data. The percentage distribution of responses for each item was visually presented using graphs to offer a more intuitive understanding of the responses. To assess the extent of PCPs' engagement in medication decision-making practices, responses to each item were scored as follows: "always" = 2 points, "occasionally" = 1 point, and "rarely" = 0 points. The total score for each participant was then calculated by summing the individual item scores, providing a composite measure of engagement. This scoring method allowed us to quantify and compare the level of engagement in medication decision-making across participants.

To explore factors associated with PCPs' engagement of medication decision-making practices, we performed multivariate linear regression analysis. Prior to the analysis, we assessed the data for linearity, normality, independence, and homoscedasticity to ensure that the assumptions for linear regression were met. This included testing for multicollinearity among independent variables and confirming that the residuals of the regression model were normally distributed [26]. In the regression model, the total score of PCPs' engagement in medication decision-making was used as the dependent variable. Independent variables such as gender, age, professional title, and educational background were included as potential predictors, while controlling for possible confounders. All statistical analyses were conducted using SPSS (version 27.0, https://www.ibm.com/support/pages/downloading-ibm-spss-statistics-27), with statistical significance set at $p < 0.05$.

## Results

### Respondent characteristics and work-related information

The electronic questionnaires were distributed to 386 PCPs, and 363 PCPs (response rate: 94%) responded and agreed to participate in the study. However, 17 questionnaires were excluded due to invalid responses, leaving 346 valid questionnaires for data analysis. The specific questionnaire data can be found in S2 File. Among the respondents, the majority (n = 180, 52.0%) were male, with a mean age of 37.8 years (range: 24–59). The majority of participants had fewer than 15 years of practice experience (n = 217, 62.7%), with most identifying as resident physicians or attending physicians (n = 272, 78.7%), and had obtained a bachelor's degree (n = 284, 82.1%) (Table 1). The demographic profile of our sample was largely consistent with the distribution patterns reported for primary care physicians in national health statistics, supporting its representativeness to a certain extent [27].

Regarding work-related characteristics, 63.9% (n = 221) of PCPs had completed standardized residency training, and 71.7% (n = 248) were responsible for family doctor contract services. Two-thirds (n = 233, 67.3%) practiced in rural PHC settings. Approximately a quarter reported a daily outpatient volume exceeding 50 patients. In terms of institutional support, 64.5% (n = 223) had participated in pharmaceutical training in the past year, 75.4% (n = 261) reported routine prescription reviews in their facilities, and 82.7% (n = 286) indicated access to electronic drug information systems. When facing challenges in medication decision-making, 42.2% of PCPs (n = 146) reported collaborating with pharmacists.

### PCPs' engagement of medication decision-making for patients with multimorbidity

Based on the 20-item questionnaire (total possible score: 40), the 346 PCPs had a mean engagement score of 31.8 (ranging from 21 to 38). The majority (n = 227, 65.6%) scored between 31 and 35 points. All participants scored above half of the total possible score.

Before initiating treatment plans, 72.5% of PCPs (n = 251) reported "always" reviewing patients' medical and medication histories, while 68.2% (n = 236) explored the purpose of the visit, and 70.2% (n = 243) addressed patients' prioritized needs. However, over one-third of respondents "occasionally" or "rarely" evaluated medication adherence and burden

**Table 1. Participant characteristics and environmental determinants (n = 346).**

| Characteristics | Category | n (%) |
|---|---|---|
| Gender | Male | 180 (52.0) |
| | Female | 166 (48.0) |
| Age, y | ≤25 | 24 (6.9) |
| | 26-35 | 150 (43.4) |
| | 36-45 | 85 (24.6) |
| | 46-55 | 74 (21.4) |
| | ≥56 | 13 (3.8) |
| Years of practice, y | ≤5 | 87 (25.1) |
| | 6-10 | 72 (20.8) |
| | 11-15 | 58 (16.8) |
| | 16-20 | 48 (13.9) |
| | 21-25 | 24 (6.9) |
| | ≥26 | 57 (16.5) |
| Professional title | Resident physician | 159 (46.0) |
| | Attending physician | 113 (32.7) |
| | Associate chief physician | 49 (14.2) |
| | Chief physician | 25 (7.2) |
| Education background | Below the bachelor | 59 (17.1) |
| | Bachelor | 284 (82.1) |
| | Master or doctor | 3 (0.9) |
| Completed the standardized residency training | Yes | 221 (63.9) |
| Family doctor contract services | Yes | 248 (71.7) |
| Location of PHC facility | rural | 233 (67.3) |
| | urban | 113 (32.7) |
| Number of outpatient visits per day | ≤50 | 258 (74.6) |
| | 51-100 | 69 (19.9) |
| | 101-150 | 7 (2.0) |
| | ≥151 | 12 (3.5) |
| Received pharmaceutical training in the past year | Yes | 223 (64.5) |
| The facility regularly conducts prescription reviews | Yes | 261 (75.4) |
| Available CDSS for medication inquiry | Yes | 286 (82.7) |
| Collaborate with pharmacists to resolve medication-related problems | Yes | 146 (42.2) |

(n = 141, 40.8%), assessed the appropriateness and effectiveness of original therapies (n = 127, 36.7%), or screened for potential drug-drug and drug-disease interactions (n = 131, 37.8%).

During treatment discussions, a majority of PCPs reported "always" eliciting patients' expectations (n = 212, 61.3%), offering treatment alternatives (n = 219, 63.3%), and communicating both benefits and feasibility (n = 198, 57.2%) as well as possible risks (n = 199, 57.5%). About 61.0% (n = 211) informed patients about how to manage adverse drug reactions, and 58.4% (n = 202) reported establishing mutually agreed treatment goals. Nonetheless, 52.5% (n = 182) reported "occasionally" or "rarely" exploring patients' preferences for medication therapy, and 47.1% (n = 163) did not consistently encourage patients to participate in the decision-making process.

Following the formulation of a treatment plan, most PCPs "always" documented medication details (n = 222, 64.2%), provided usage instructions (n = 240, 69.4%), discussed precautions (n = 232, 67.1%), and arranged

follow-up visits (n = 203, 58.7%). However, 39.6% (n = 137) "occasionally" or "rarely" informed patients about common adverse drug reactions, and 5.5% (n = 19) reported rarely asking if patients had further questions about their medications (Fig 1).

## Perceived difficulties in making medication decisions

A substantial proportion of PCPs reported facing various challenges in medication decision-making. Specifically, 82.8% (n = 286) "strongly agreed" or "agreed" that they lacked adequate knowledge regarding drug use, while 70.8% (n = 245) acknowledged difficulties in discussing alternative medication options with patients. Additionally, 61.6% (n = 213) reported struggling to balance the benefits and risks of medication therapies, and over half found it challenging to explain the potential adverse outcomes (n = 212, 61.3%) and identify drug-drug or drug-disease interactions (n = 192, 55.5%). Nevertheless, more than two-thirds of respondents (n = 235) "strongly disagreed" or "disagreed" that clinical inertia prevented them from making appropriate medication decisions (Fig 2).

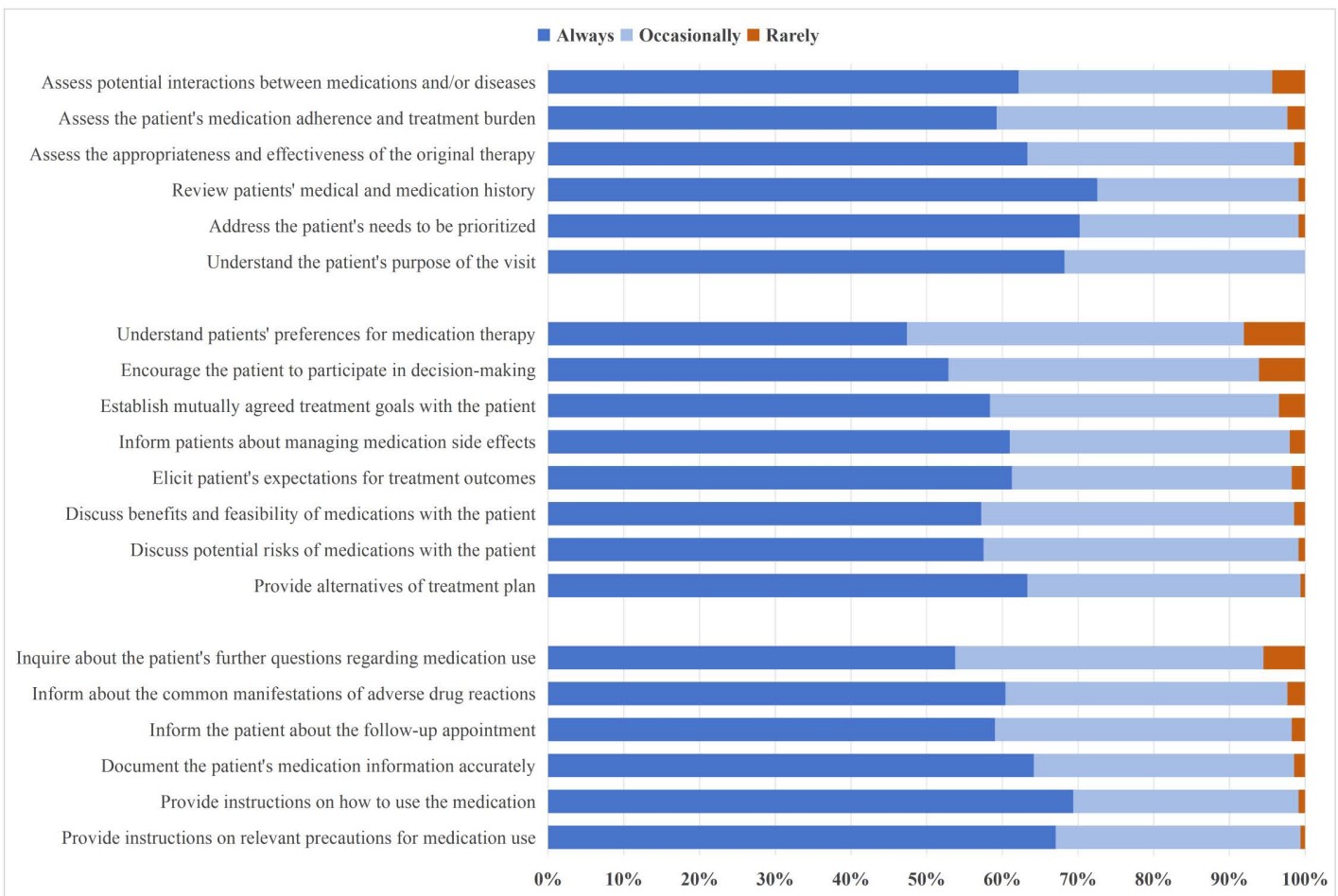

**Fig 1. Frequency of primary care physicians' engagement in medication decision-making for multimorbidity.** This figure displays the percentage distribution of responses for 20 items assessing PCPs' self-reported frequency of performing specific medication decision-making tasks for patients with multimorbidity during the past month. Response options were "always", "occasionally", or "rarely".

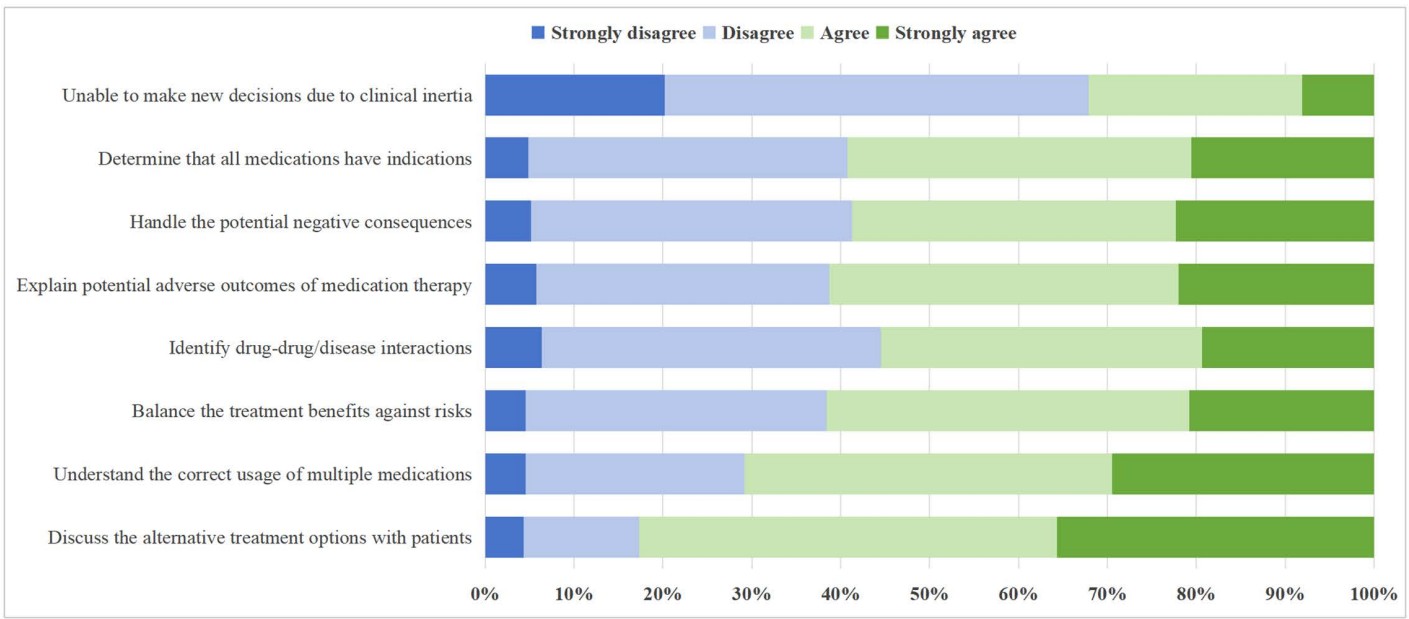

**Fig 2. Perceived difficulties in medication decision-making for multimorbidity among primary care physicians.** This figure displays the percentage distribution of responses for eight items assessing PCPs' perceived difficulties in making medication decisions for patients with multimorbidity. Respondents rated their agreement on a four-point Likert scale ranging from "strongly disagree" to "strongly agree".

### Factors associated with PCPs' engagement of medication decision-making

Multivariate linear regression identified several factors associated with higher levels of PCPs' overall engagement in medication decision-making for multimorbidity (Table 2). Female physicians demonstrated significantly greater engagement (B = 2.245, 95% CI: 0.102 to 4.387), as did those with higher professional titles (B = 1.700, 95% CI: 0.121 to 3.278). Work-related characteristics also played a key role. PCPs who had received pharmaceutical training in the past year were more engaged in decision-making (B = 5.764, 95% CI: 1.559 to 9.969), as were those who reported collaborating with pharmacists to resolve medication-related problems (B = 7.048, 95% CI: 3.313 to 10.782). In contrast, a higher outpatient volume was negatively associated with engagement (B = −0.046, 95% CI: −0.083 to −0.008), indicating that time constraints may hinder comprehensive medication management.

### Discussion

This cross-sectional study investigated how PCPs in five regions of Zhejiang province, China, engage in medication decision-making for patients with multimorbidity. We comprehensively examined PCPs' self-reported practices across different stages of the consultation process and the perceived barriers they encounter when making medication decisions. Furthermore, using multivariate linear regression analysis, we identified three work-related factors that are significantly associated with the overall level of PCPs' engagement in medication decision-making for patients with multimorbidity.

A noteworthy finding of this study is that more than one-third of PCPs reported not frequently evaluating the appropriateness of existing therapies, assessing medication adherence, or screening for potential drug interactions. This is particularly concerning, as patients with multimorbidity are at significantly increased risk of medication-related harm. Prior research has demonstrated that failures in medication reconciliation and inadequate screening for drug interactions are among the leading causes of preventable adverse drug events in primary care [28]. The work-related factors identified in our study, such as limited access to decision-support resources, insufficient training in polypharmacy management, and

**Table 2. Factors associated with PCPs' overall engagement of medication decision-making.**

| Model | Unstandardized Coefficients | | Standardized Coefficients | t | Sig. | 95% Confidence Interval | |
|---|---|---|---|---|---|---|---|
| | B | Std. Error | Beta | | | Lower Bound | Upper Bound |
| Gender | 2.245 | 1.088 | 0.127 | 2.063 | 0.040 | 0.102 | 4.387 |
| Age | −0.098 | 0.157 | −0.108 | −0.628 | 0.531 | −0.407 | 0.210 |
| Years of practice | 0.003 | 0.158 | 0.004 | 0.020 | 0.984 | −0.308 | 0.314 |
| Professional title | 1.700 | 0.802 | 0.180 | 2.120 | 0.035 | 0.121 | 3.278 |
| Education background | 2.374 | 1.437 | 0.110 | 1.652 | 0.100 | −0.456 | 5.205 |
| Completed standardized residency training | −0.363 | 1.165 | −0.020 | −0.312 | 0.756 | −2.658 | 1.932 |
| Family doctor contract services | −2.246 | 1.592 | −0.117 | −1.410 | 0.160 | −5.382 | 0.890 |
| Location of PHC facility | −1.442 | 1.319 | −0.079 | −1.094 | 0.275 | −4.039 | 1.154 |
| Number of outpatient visits per day | −0.046 | 0.018 | −0.471 | −2.507 | 0.020 | −0.083 | −0.008 |
| Received pharmaceutical training in the past year | 5.764 | 2.135 | 0.161 | 2.699 | 0.007 | 1.559 | 9.969 |
| The facility regularly conducts prescription reviews | 4.850 | 2.789 | 0.111 | 1.739 | 0.083 | −0.642 | 10.341 |
| Available CDSS for medication inquiry | 1.854 | 1.369 | 0.086 | 1.354 | 0.177 | −0.842 | 4.550 |
| Collaborate with pharmacists to resolve medication-related problems | 7.048 | 1.896 | 0.222 | 3.716 | 0.001 | 3.313 | 10.782 |

time pressures during consultations, align with barriers previously reported in both high-income and resource-constrained countries [12,13,29,30]. These findings suggest that systemic and educational interventions targeting these obstacles may be necessary to optimize medication safety for patients with complex care needs.

Our findings indicate that nearly half of the surveyed PCPs did not consistently understand patients' medication preferences or actively involve them in the decision-making process. This stands in contrast to current clinical guidelines, which strongly advocate for patient-centered care and shared decision-making, particularly in the management of multimorbidity [31]. Evidence from international studies has shown that shared decision-making not only improves medication adherence and patient satisfaction but also may have a positive association with clinical outcomes in multimorbid populations [31]. The suboptimal implementation observed in our study may be attributable to several contextual factors, including cultural norms that prioritize physician authority, time constraints in busy primary care settings, and insufficient training in communication and shared decision-making techniques. Moreover, research has demonstrated that engaging patients in decision-making processes enhances their understanding of treatment plans and reduces decisional conflict and subsequent regret [32,33]. Addressing these barriers through targeted educational and system-level interventions may be essential for promoting more collaborative and effective care in the primary care setting.

The high proportion of PCPs reporting insufficient pharmacological knowledge and difficulty in conducting risk-benefit assessments underscores a significant gap in clinical training. This finding aligns with a well-documented limitation in current practice: most clinical guidelines remain disease-specific and fail to account for the complexity of managing patients with multiple coexisting conditions [34]. As a result, PCPs are often left without clear, evidence-based strategies for prioritizing treatment goals and balancing therapeutic trade-offs in multimorbidity care [35]. The absence of integrated, multimorbidity-oriented clinical guidelines, alongside a paucity of robust evidence regarding the benefits and harms of combined therapies, poses a considerable challenge to individualized, rational prescribing in primary care settings [34]. Interestingly, our findings show that clinical inertia was not perceived as a major barrier by most respondents. This suggests that limitations in pharmacological knowledge and systemic constraints, such as inadequate continuing education and lack of clinical decision support, may be more prominent contributors to suboptimal prescribing than physician unwillingness to modify existing treatment regimens. This is consistent with previous studies highlighting that confidence and competence, rather than attitude alone, often dictate prescriber behavior in complex cases [36,37].

Our regression analysis identified several key factors associated with increased engagement in medication decision-making. Notably, participation in pharmaceutical training was positively linked to greater engagement, underscoring the importance of ongoing professional development in addressing knowledge gaps. Collaboration with pharmacists emerged as the strongest predictor, highlighting the growing evidence for interdisciplinary care models in managing complex medication regimens [38]. Conversely, higher daily patient volume was negatively associated with engagement, providing further evidence that time pressures significantly hinder the ability to engage in thoughtful, individualized prescribing [29]. These findings emphasize the urgent need for system-level reforms that reduce workload burdens and support the delivery of high-quality medication management for multimorbid patients.

The findings of this study highlight several critical areas for future investigation and practical improvement. First, the identified knowledge gaps and difficulties in risk-benefit assessment highlight the need for targeted educational interventions, particularly in pharmacology and multimorbidity management, to strengthen PCPs' clinical decision-making capabilities. Second, given the strong association between pharmacist collaboration and engagement in medication decision-making, future practice should prioritize the integration of pharmacists into primary care teams, while research should explore effective models of interdisciplinary collaboration. Third, the negative impact of high patient volume on engagement underscores the importance of addressing workload through system-level innovations (such as team-based care, task shifting, or digital support tools) to ensure that time constraints do not compromise medication safety. Finally, the limited incorporation of patient preferences suggests a need to enhance communication training and promote shared decision-making frameworks in routine practice, enabling more individualized and patient-centered care.

This study has several limitations. First, the sample was limited to one province in China, which may affect the generalizability of the findings to other regions. Zhejiang, as an economically developed province in eastern China, has relatively well-trained PCPs and more resource-rich healthcare settings compared to less developed or remote areas. Nevertheless, we included participants from both urban and rural (PHC facilities to enhance representativeness and capture a range of practice environments. Furthermore, the high proportion of resident physicians (46%) in our sample, who typically have less experience and lower involvement in complex decision-making, may limit the generalizability of the findings to the broader population. Third, some respondents may not have fully expressed their perspectives due to time constraints, recall bias, or cognitive limitations, which could have influenced the accuracy of the self-reported data. Besides, there is a possibility of social desirability bias, as the survey was administered by local health authorities. Although we clearly communicated confidentiality measures and emphasized anonymity in the informed consent process, some participants may still have responded in ways they perceived as socially or professionally favorable.

## Conclusion

While PCPs reported a general level of engagement in medication decision-making practices, significant gaps exist in the management of complex therapies and the implementation of patient-centered care. Challenges related to limited pharmacological knowledge, insufficient communication skills, and low levels of shared decision-making were commonly reported. To address these problems, future efforts should prioritize the development of targeted pharmaceutical training and the promotion of interprofessional collaboration, particularly between physicians and pharmacists, to enhance PCPs' capacity to manage multimorbidity more effectively and deliver individualized, high-quality care.

## Supporting information

**S1 File. A survey on primary care physicians' engagement in medication decision-making for patients with multimorbidity.**
(DOCX)

**S2 File. PCP_MedDecision_Multimorbidity_Survey_Data. The original data and statistics of the questionnaire survey on Engagement of Primary Care Physicians in Medication Decision-making for Patients with Multimorbidity in China: A Cross-sectional Study.**
(PDF)

## Acknowledgments

The authors thank all the physicians who participated in the survey.

## Author contributions

**Conceptualization:** Liyan Han, Leyi Jiang, Yuling Tong, Zhijie Xu.

**Data curation:** Liyan Han, Yu Xia, Lingyan Wu, Zhijie Xu.

**Formal analysis:** Liyan Han, Zhijie Xu.

**Investigation:** Liyan Han, Lingyan Wu.

**Methodology:** Leyi Jiang, Zhijie Xu.

**Project administration:** Lingyan Wu, Yi Guo, Yuling Tong.

**Resources:** Lingyan Wu, Yuling Tong.

**Supervision:** Yi Guo.

**Validation:** Leyi Jiang, Mi Yao, Ming Yan, Junhai Zhen.

**Visualization:** Liyan Han, Zhijie Xu.

**Writing – original draft:** Liyan Han, Leyi Jiang.

**Writing – review & editing:** Zhijie Xu.

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
