## [Decision Letter · Decision Letter 0]

27 Jan 2026

Dear Dr. Xu,

Thank you for submitting your manuscript to PLOS ONE. After careful consideration, we feel that it has merit but does not fully meet PLOS ONE’s publication criteria as it currently stands. Therefore, we invite you to submit a revised version of the manuscript that addresses the points raised during the review process.

We look forward to receiving your revised manuscript.

Kind regards,

Pedro Kallas Curiati, M.D., Ph.D.

Academic Editor

PLOS One

Journal Requirements:

3. In the online submission form, you indicated that due to privacy and confidentiality restrictions under Chinese regulations and the conditions of participant consent, the full dataset cannot be made publicly available. De-identified data can be provided to qualified researchers upon reasonable request, subject to a formal data use agreement. Requests should be directed to the corresponding author.

Reviewers' comments:

Reviewer's Responses to Questions

**Comments to the Author**

1. Is the manuscript technically sound, and do the data support the conclusions?

Reviewer #1: Yes

Reviewer #2: Yes

2. Has the statistical analysis been performed appropriately and rigorously?

Reviewer #1: Yes

Reviewer #2: Yes

3. Have the authors made all data underlying the findings in their manuscript fully available?

Reviewer #1: Yes

Reviewer #2: No

4. Is the manuscript presented in an intelligible fashion and written in standard English?

Reviewer #1: Yes

Reviewer #2: Yes

Reviewer #1: Thank you for allowing me to review this manuscript. The manuscript describes a cross-sectional evaluation of PCPs' medication practices during clinic visits. The manuscript is generally well-written. The statistical analysis is adequate. The study seems to align with the journal's readership. Focusing on PCPs is important due to their role in the health care system. The study seems to be a worthy addition to the literature.

The reporting could be slightly improved to enhance the clarity of the article. Please find my comments immediately below.

Title: Good

Abstract: Good

Introduction

-Line 31: Good description of PHC in China. What is the role of specialist physicians regarding medication prescribing? In our research we found that PCPs feel very uncomfortable adjusting therapies or deprescribing medications that they did not themselves prescribe. How does the PCP-specialist physician relationship impact medication management?

Methods

-Line 82: Thank you for providing the survey. Again, lack of inclusion about items related to adjusting therapies that the provider itself did not start is a concern since the authors conducted a literature review.

-Line 101: Please cite a reference for the final statement in this paragraph.

-Lines 108-109: Why was never not included in the likert type scale since Always is included.

Results

-Overall: Please provide n's where you have percentages.

-Line 178: Revise from "...nearly half..." to the majority.

-Line 180: Revise from "...junior or intermediate professional titles..." to were resident physicians or attending physicians.

-Line 182: Would be helpful to know how representative the sample is to the Chinese PCP population. Please revise the Methods to include some statistical comparisons and present the findings here.

-Line 183: Is it common for PCPs not to have completed any residency training? Also, is it possible that there might be some physicians who were pharmacists or had some specialized pharmacology training prior to becoming a PCP?

-Line 228: Please provide descriptive statistics for the "score" so readers can better understand where the PCPs landed in terms of medication practices.

Discussion:

Line numbering stops here.

-The sentence "...align with barriers previously reported in both..." needs to be finished.

-Page 13, the sentence "...currently clinical guidelines which strongly advocate..." needs to be cited. Preferably one about polypharmacy or multimorbidity management.

-Page 14, the sentence "...research has demonstrated that engaging patients in decision-making processes...", has research found SDM improves medication adherence and/or disease outcomes?

-Page 14, the sentence "...aligns with a well-documented limitation in current practice:..." needs to be cited.

Reviewer #2: I am grateful for the opportunity to review this paper.

The work is very interesting and discuss an important topic related to pharmacotherapy management and drug safety. Furthermore, it is very well written.

Below are some comments on the paper:

1. The first cutoff point reported for the “number of outpatient visits per day” is high (≤50 patients per day). This seems quite striking. Considering that the question was free text (open question), I suggest exploring others range, or explaining how and why this cutoff was defined.

2. Please revise the statement “one-third reported a daily outpatient volume exceeding 50 patients” (line 186) to “a quarter reported…”, as this corresponds to approximately 25%.

3. The proportion of resident physicians represents almost half (46%) of all respondents. This is a significative characteristic of the sample, and its impact on the results should be discussed.

4. Regarding the highest academic degree, it is not clear in the manuscript what is meant by “college” and “bachelor” within the context of Chinese medical education and training, especially given international differences in educational systems.

**Do you want your identity to be public for this peer review?** For information about this choice, including consent withdrawal, please see our For information about this choice, including consent withdrawal, please see our Privacy Policy .

Reviewer #1: No

Reviewer #2: No

You may also use PLOS’s free figure tool, NAAS, to help you prepare publication quality figures: https://journals.plos.org/plosone/s/figures#loc-tools-for-figure-preparation

---

## [Author Response · Author response to Decision Letter 1]

19 Feb 2026

Responses to the reviewer 1

【COMMENT 1】Good description of PHC in China. What is the role of specialist physicians regarding medication prescribing? In our research we found that PCPs feel very uncomfortable adjusting therapies or deprescribing medications that they did not themselves prescribe. How does the PCP-specialist physician relationship impact medication management?

【RESPONSE 1】Thank you for raising this important point regarding the relationship between PCPs and specialists in medication management. We have revised the Introduction to explicitly address this dynamic, noting that clinical inertia often stems from PCPs’ hesitancy to adjust prescriptions initiated by specialists. This addition further clarifies a key contextual barrier in our study setting.

(Introduction) Numerous barriers hinder their ability to optimize treatment, including clinical inertia, patient resistance to changes in medication regimens, limited access to decision support tools, inadequate training in managing polypharmacy, and time constraints during consultations [12,13]. Specifically, clinical inertia among PCPs often stems from their hesitancy to modify specialist-initiated regimens due to deference to authority, uncertainty about prescribing rationale, and legal concerns, thus compromising medication safety and care quality [14]. Moreover, the lack of clinical guidelines and insufficient evidence for treating patients with multimorbidity make it difficult to formulate individualized medication plans [15].

References added:

[14] Phillips LS, Branch WT, Cook CB, Doyle JP, El-Kebbi IM, Gallina DL, et al. Clinical inertia. Ann Intern Med. 2001; 135(9):825-34. doi: 10.7326/0003-4819-135-9-200111060-00012.

【COMMENT 2】Thank you for providing the survey. Again, lack of inclusion about items related to adjusting therapies that the provider itself did not start is a concern since the authors conducted a literature review.

【RESPONSE 2】Thank you for raising this important point regarding the scope of our survey. We wish to clarify that the questionnaire items were designed to assess PCPs’ decision-making engagement in a broad clinical context, specifically, when initiating a new prescription or modifying an existing medication regimen for a patient, irrespective of whether the original therapy was started by the PCP or another provider (e.g., a specialist). Thus, the instrument captures participatory decision-making practices across both self-initiated and externally initiated therapies under the PCP’s ongoing management.

【COMMENT 3】Manuscript line 101: Please cite a reference for the final statement in this paragraph.

【RESPONSE 3】Thank you for your suggestion regarding the citation for the psychometric properties. We would like to clarify that the reported metrics (Cronbach’sα, ICC, and CFI) were derived from the pilot testing phase of this study, which was conducted to assess the psychometric properties of the self-developed questionnaire. This preliminary validation confirms the instrument's robustness for the main survey. In line with your advice, we have supplemented the relevant methodological literature on psychometric evaluation in this section.

(Methods－Instrument) The instrument demonstrated good internal consistency (Cronbach’s α= 0.87 for the full scale), strong test–retest reliability (intraclass correlation coefficient = 0.84 over a two-week interval) and satisfactory construct validity (χ²/df=1.98, CFI=0.93, TLI=0.91, RMSEA=0.06) [18].

References:

[18] Kline RB. Principles and practice of structural equation modeling, 3rd ed. New York: Guilford Publications, 2011.

【COMMENT 4】Why was NEVER not included in the likert type scale since ALWAYS is included.

【RESPONSE 4】We sincerely appreciate the reviewer’s insightful question regarding the response options in our questionnaire. Our decision to use a three-point scale ("always," "occasionally," "rarely") was based on methodological and practical considerations. We opted against including a "never" option to minimize potential social desirability bias and respondent discomfort, as an absolute term might be perceived as an admission of not performing a core professional activity. The term “rarely” was chosen to capture low-frequency behaviors while maintaining a constructive and less prescriptive tone. Furthermore, from an analytical perspective, we anticipated that a “never” response would be selected infrequently, potentially creating a highly skewed distribution that could complicate subsequent statistical analyses. The chosen scale provided a balanced distribution suitable for our analysis while maintaining a constructive response tone.

【COMMENT 5】Please provide n’s where you have percentages.

【RESPONSE 5】Thank you for this helpful suggestion. We have revised the manuscript to ensure that all percentages reported in the Results section are now accompanied by their corresponding absolute numbers (n). As the changes are made throughout the relevant sections, we have not listed them individually here but have implemented them comprehensively in the revised manuscript to enhance the clarity and transparency of our data presentation.

(Results－Respondent Characteristics and Work-related Information)

Regarding work-related characteristics, 63.9% (n=221) of PCPs had completed standardized residency training, and 71.7% (n=248) were responsible for family doctor contract services. Two-thirds (n=233, 67.3%) practiced in rural PHC settings. Approximately a quarter reported a daily outpatient volume exceeding 50 patients. In terms of institutional support, 64.5% (n=223) had participated in pharmaceutical training in the past year, 75.4% (n=261) reported routine prescription reviews in their facilities, and 82.7% (n=286) indicated access to electronic drug information systems. When facing challenges in medication decision-making, 42.2% of PCPs (n=146) reported collaborating with pharmacists.

【COMMENT 6 AND 7】Revise from “...nearly half...” to the majority. Revise from "...junior or intermediate professional titles..." to were resident physicians or attending physicians.

【RESPONSE 6 AND 7】Thank you for your kind reminder. We have carefully revised the wording to ensure accuracy and clarity in the manuscript. The text has been updated as suggested, specifically changing “nearly half” to “the majority” and revising “held junior or intermediate professional titles” to “were resident physicians or attending physicians”.

(Result－Respondent Characteristics and Work-related Information) Among the respondents, the majority (n=180, 52.0%) were male, with a mean age of 37.8 years (range: 24–59). The majority of participants had fewer than 15 years of practice experience (n=217, 62.7%), with most identifying as resident physicians or attending physicians (n=272, 78.7%), and had obtained a bachelor’s degree (n=284, 82.1%) (Table 1).

【COMMENT 8】Would be helpful to know how representative the sample is to the Chinese PCP population. Please revise the Methods to include some statistical comparisons and present the findings here.

【RESPONSE 8】Thank you for this valuable suggestion regarding the representativeness of our sample. In response, we have added a concise comparative statement in the Results section (following the presentation of participants’ demographic and practice characteristics). By referencing the latest national health statistics (e.g., China Health Statistics Yearbook 2024), we note that the gender, age, educational, and professional title distribution of our sample aligns closely with the documented profile of physicians in China—where, for instance, 52.1% are male, 35.2% are aged 35–44, and 78.4% hold a bachelor’s degree or higher. This addition references national statistical data and indicates that the profile of our sample aligns broadly with that of the primary care physician population in China, thereby supporting the reasonable representativeness of our cohort within the studied context.

(Result－Respondent Characteristics and Work-related Information) The majority of participants had fewer than 15 years of practice experience (n=217, 62.7%), with most identifying as resident physicians or attending physicians (n=272, 78.7%), and had obtained a bachelor’s degree (n=284, 82.1%) (Table 1). The demographic profile of our sample was largely consistent with the distribution patterns reported for primary care physicians in national health statistics, supporting its representativeness to a certain extent [27].

References added:

[27] National Health Commission of China. 2024 China Health Statistics Yearbook. Beijing: Peking Union Medical College Press, 2025.

【COMMENT 9】Is it common for PCPs not to have completed any residency training? Also, is it possible that there might be some physicians who were pharmacists or had some specialized pharmacology training prior to becoming a PCP?

【RESPONSE 9】Thank you for your questions regarding the training background of physicians in China. We would like to provide a detailed explanation regarding this matter. Approximately a decade ago, China introduced a three-year standardized residency training program, which is now generally required for medical graduates to obtain a training completion certificate before practicing independently. Regarding pharmacology training, all PCPs in China are graduates of clinical medicine programs, while pharmacists cannot transition to become physicians. The pharmacology education these physicians receive during both undergraduate study and residency training is fragmented and limited, lacking the depth of specialized pharmacy training.

【COMMENT 10】Please provide descriptive statistics for the “score” so readers can better understand where the PCPs landed in terms of medication practices.

【RESPONSE 10】Thank you for this valuable comment. We agree that descriptive statistics regarding the overall engagement scores would help readers better interpret the level of PCPs' involvement in medication decision-making. We acknowledge that this information was initially missing from the manuscript.

In response to your suggestion, we have now added a paragraph under Results section (PCPs’ Engagement of Medication Decision-making for Patients with Multimorbidity) to present the descriptive statistics of the engagement scores. We believe this addition provides a clearer and more comprehensive overview of PCPs’ medication decision-making practices.

(Results－PCPs’ Engagement of Medication Decision-making for Patients with Multimorbidity) Based on the 20-item questionnaire (total possible score: 40), the 346 PCPs had a mean engagement score of 31.8 (ranging from 21 to 38). The majority (n=227, 65.6%) scored between 31 and 35 points. All participants scored above half of the total possible score.

Before initiating treatment plans, 72.5% of PCPs (n=251) reported “always” reviewing patients’ medical and medication histories, while 68.2% (n=236) explored the purpose of the visit, and 70.2% (n=243) addressed patients’ prioritized needs.

【COMMENT 11】The sentence “...align with barriers previously reported in both...” needs to be finished.

【RESPONSE 11】Thank you for your kind reminder. We have carefully revised the wording to ensure the completeness and clarity of the sentence. The incomplete statement in the original text has been corrected by adding the missing word, specifically amending “resource-constrained” to “resource-constrained countries”.

(Discussion) The work-related factors identified in our study, such as limited access to decision-support resources, insufficient training in polypharmacy management, and time pressures during consultations, align with barriers previously reported in both high-income and resource-constrained countries [12,13,29,30].

【COMMENT 12】The sentence “...currently clinical guidelines which strongly advocate...” needs to be cited. Preferably one about polypharmacy or multimorbidity management.

【RESPONSE 12】Thank you for your thoughtful comment regarding the need to substantiate the statement about clinical guidelines. We acknowledge that the citation in our initial submission could have been more explicit. Upon review, we confirm that the recommended reference on polypharmacy and multimorbidity management (Muth et al., 2019) was already included in our manuscript’s reference list. To strengthen the support for this specific claim, we have now explicitly cited this guideline at the relevant point in the sentence.

(Discussion) This stands in contrast to current clinical guidelines, which strongly advocate for patient-centered care and shared decision-making, particularly in the management of multimorbidity [31].

References:

[31] Muth C, Blom JW, Smith SM, Johnell K, Gonzalez-Gonzalez AI, Nguyen TS, et al. Evidence supporting the best clinical management of patients with multimorbidity and polypharmacy: a systematic guideline review and expert consensus. J Intern Med. 2019, 285(3):272-288. doi: 10.1111/joim.12842.

【COMMENT 13】The sentence "...research has demonstrated that engaging patients in decision-making processes...", has research found SDM improves medication adherence and/or disease outcomes?

【RESPONSE 13】Thank you for your valuable suggestion regarding the need to substantiate the link between SDM and patient outcomes. While our originally cited reference (Laviana et al., 2017) does state that involved patients “have better outcomes”, we acknowledge that this correlational language does not constitute direct, quantified empirical data on specific clinical endpoints. To more accurately reflect the available evidence and enhance the academic rigor of our argument, we have revised the relevant sentence. We have changed the phrasing from “...leads to better clinical outcomes...” to the more measured “...may have a positive association with clinical outcomes...” This modification aims to precisely convey the supportive yet indicative relationship suggested by the literature, rather than asserting a definitive causal link. We appreciate your guidance in prompting this clarification and are open to any further suggestions.

(Discussion) Evidence from international studies has shown that shared decision-making not only improves medication adherence and patient satisfaction but also may have a positive association with clinical outcomes in multimorbid populations [31].

【COMMENT 14】The sentence “...aligns with a well-documented limitation in current practice:...” needs to be cited.

【RESPONSE 14】Thank you for your thoughtful comment regarding the need to substantiate the statement about the well-documented limitation in current practice. We acknowledge that the citation in our initial submission could have been more explicit. Upon review, we confirm that the recommended reference (Wang et al., Ageing Res Rev, 2024), which systematically reviews the limitations of current disease-specific guidelines in addressing multimorbidity, was already included in our manuscript's reference list. To directly address your point and strengthen the support for this specific claim, we have now explicitly cited this review at the relevant point in the sentence.

(Discussion) This finding aligns with a well-documented limitation in current practice: most clinical guidelines remain disease-specific and fail to account for the complexity of managing patients with multiple coexisting conditions [34].

References:

[34] Wang Z, Zhu D, Zhang H, Wang L, He H, Li Z, et al. Recommendations and quality of multimorbidity guidelines: A systematic review. Ageing Res Rev. 2024; 102:102559. DOI: 10.1016/j.arr.2024.102559.

Responses to the reviewer 2

【COMMENT 1】The first cutoff point reported for the “number of outpatient visits per day” is high (≤50 patients per day). This seems quite striking. Considering that the question was free text (open question), I suggest exploring others range, or explaining how and why this cutoff was defined.

【RESPONSE 1】Thank you for your insightful comment and for raising this important point regarding the cutoff values for the daily number of outpatient visits. We appreciate your careful review. In our survey, which used an open-ended question, the reported daily patient volumes varied considerably, ranging from single digits to over 150 visits per day.

---

## [Editor Report · Decision Letter 1]

22 Feb 2026

Engagement of Primary Care Physicians in Medication Decision-making for Patients with Multimorbidity in China: A Cross-sectional Study

PONE-D-25-67618R1

Dear Dr. Xu,

We’re pleased to inform you that your manuscript has been judged scientifically suitable for publication and will be formally accepted for publication once it meets all outstanding technical requirements.

Kind regards,

Pedro Kallas Curiati, M.D., Ph.D.

Academic Editor

PLOS One

---

## [Editor Report · Acceptance letter]

PONE-D-25-67618R1

PLOS One

Dear Dr. Xu,

I'm pleased to inform you that your manuscript has been deemed suitable for publication in PLOS One. Congratulations! Your manuscript is now being handed over to our production team.

Kind regards,

on behalf of

Dr. Pedro Kallas Curiati

Academic Editor

PLOS One